

# Random sampling causes the low reproducibility of rare eukaryotic OTUs in Illumina COI metabarcoding

Matthieu Leray[1,2] and Nancy Knowlton[1]

[1] National Museum of Natural History, Smithsonian Institution, Washington, D.C., USA
[2] Smithsonian Tropical Research Institute, Smithsonian Institution, Panama City, Balboa, Ancon, Republic of Panama

Corresponding author
Matthieu Leray,
leray.upmc@gmail.com

## ABSTRACT

DNA metabarcoding, the PCR-based profiling of natural communities, is becoming the method of choice for biodiversity monitoring because it circumvents some of the limitations inherent to traditional ecological surveys. However, potential sources of bias that can affect the reproducibility of this method remain to be quantified. The interpretation of differences in patterns of sequence abundance and the ecological relevance of rare sequences remain particularly uncertain. Here we used one artificial mock community to explore the significance of abundance patterns and disentangle the effects of two potential biases on data reproducibility: indexed PCR primers and random sampling during Illumina MiSeq sequencing. We amplified a short fragment of the mitochondrial Cytochrome c Oxidase Subunit I (COI) for a single mock sample containing equimolar amounts of total genomic DNA from 34 marine invertebrates belonging to six phyla. We used seven indexed broad-range primers and sequenced the resulting library on two consecutive Illumina MiSeq runs. The total number of Operational Taxonomic Units (OTUs) was ∼4 times higher than expected based on the composition of the mock sample. Moreover, the total number of reads for the 34 components of the mock sample differed by up to three orders of magnitude. However, 79 out of 86 of the unexpected OTUs were represented by <10 sequences that did not appear consistently across replicates. Our data suggest that random sampling of rare OTUs (e.g., small associated fauna such as parasites) accounted for most of variation in OTU presence–absence, whereas biases associated with indexed PCRs accounted for a larger amount of variation in relative abundance patterns. These results suggest that random sampling during sequencing leads to the low reproducibility of rare OTUs. We suggest that the strategy for handling rare OTUs should depend on the objectives of the study. Systematic removal of rare OTUs may avoid inflating diversity based on common $\beta$ descriptors but will exclude positive records of taxa that are functionally important. Our results further reinforce the need for technical replicates (parallel PCR and sequencing from the same sample) in metabarcoding experimental designs. Data reproducibility should be determined empirically as it will depend upon the sequencing depth, the type of sample, the sequence analysis pipeline, and the number of replicates. Moreover, estimating relative biomasses or abundances based on read counts remains elusive at the OTU level.

## INTRODUCTION

High-throughput sequencing of short homologous DNA fragments from mixed bulk samples, an approach referred to as DNA metabarcoding, has become very popular for monitoring diversity because it circumvents limitations inherent to morphological identifications (*Taberlet et al., 2012*; *Ji et al., 2013*; *Lejzerowicz et al., 2015*). Despite its enormous potential, metabarcoding is also subject to some biases. Understanding how various metabarcoding methodologies affect estimates of species richness (α diversity) and patterns of community composition (β diversity) has therefore been an active area of research (*Bik et al., 2012*; *Leray & Knowlton, 2016*). For example, various empirical studies have shown that sampling techniques (*Hirai et al., 2015*), sample storage and DNA extraction methods (*Brannock & Halanych, 2015*; *Deiner et al., 2015*) induce taxon-specific biases. Others have highlighted how additional biases occur during PCR amplification as a result of differential primer binding efficiency (*Elbrecht & Leese, 2015*). Informatics procedures for data quality control (*Schloss, Gevers & Westcott, 2011*; *Rossberg, Rogers & McKane, 2014*), sequence clustering (*Brown et al., 2015*; *Flynn et al., 2015*) and taxonomically biased public repositories (e.g., GenBank) (*Guillou et al., 2013*; *Leray & Knowlton, 2016*) are now well-identified sources of variation in taxon detection. Together, these empirical studies quantifying biases have highlighted the importance of implementing rigorous experimental designs and adhering to strict laboratory protocols. They have also helped develop statistical frameworks that take into account methodological artifacts for the inference of taxon richness, biomass and community dissimilarity from sequence data (*Shelton et al., 2016*). Despite significant progress, the state of knowledge remains limited. Here, we further help to understand how metabarcoding sequence data can be used to make ecological predictions by exploring two potentially important sources of biases: indexed PCR primers and random sampling during Illumina MiSeq sequencing.

It is now well accepted that some level of primer bias is inevitable with PCR-based approaches because no primer set is truly universal (*Leray et al., 2013*). Differences in primer affinity among taxonomic groups prevent the detection of some Operational Taxonomic Units (OTUs) and result in inaccurate estimates of relative abundances, particularly within complex samples (*Pinto & Raskin, 2012*; *Deagle et al., 2013*; *Elbrecht & Leese, 2015*; *Piñol et al., 2015*). Pervasive amplification biases of some primer sets have even led some to suggest that metabarcoding should only be interpreted in terms of OTU presence-absence, not in terms of relative abundance (*Elbrecht & Leese, 2015*; *Piñol et al., 2015*). The addition of short indices to PCR primers has been suggested to induce further amplification biases. Indexed PCR primers are commonly used to minimize the per sample cost of sequencing by allowing numerous samples to be sequenced on a single run (*Binladen et al., 2007*). Unique indices are attached to the 5′ end of the forward or reverse primer (or both) during oligonucleotide synthesis. Each sample is then amplified with a unique set of indexed primers to add a "DNA identifier" to each amplified fragment, allowing reads produced by parallel sequencing to be assigned to their original sample. This approach assumes that indices differing by two to three base pairs do not modify the affinity of
primers with DNA templates during PCR cycles, an assumption that could be violated (*Berry et al., 2011*; *O'Donnell et al., 2016*), particularly for highly diverse communities.

Another much more neglected source of variation in OTU detection may be attributed solely to random sampling effects that are completely independent of any technical causes. Random sampling, or the random draw of OTUs from a pool where OTUs have equal chances of being selected, is most likely to occur during PCR amplification, library preparation (e.g., ligation of adaptors) and sequencing (regardless of the platform used), with the greatest impact on the detection of rare OTUs (*Zhan & MacIsaac, 2015*). To date, two studies that assessed the reproducibility of amplicon-based studies proposed random sampling artifacts as a significant concern for the reproducibility of presence-absence data. The first suggested that random sampling was the main cause for the low levels of overlap in OTU composition between PCR replicates of microbial communities (*Zhou et al., 2011*), but the authors used different indexed PCR primers across technical replicates, a potential confounding source of bias. The second and more recent study came to the same conclusion for metazoan communities, but they compared biological replicates (i.e., two parallel fractions of the same sample) rather than technical replicates (*Zhan et al., 2014*).

The present study was designed to explore the significance of sequence abundance patterns and to disentangle technical biases caused by the addition of short indices to PCR primers versus random sampling occurring during Illumina MiSeq sequencing. We analysed the reproducibility of Illumina MiSeq data for a single mock sample comprising DNA of 34 marine invertebrates belonging to six phyla. We targeted a short fragment of the mitochondrial Cytochrome c Oxidase Subunit I (COI) with seven indexed broad-range primers. The COI gene, used as a standard barcode marker for most metazoan groups, is highly conserved at the amino acid level, which helps with multi-sequence alignments and the subsequent detection of sequence artifacts in metabarcoding datasets. The library, composed of seven indexed amplicons, was sequenced in two consecutive Illumina MiSeq runs. We assess the consistency of presence-absence and abundance data across replicates and propose solutions to enhance the reproducibility of metabarcoding for biodiversity monitoring.

## MATERIALS AND METHODS

### Preparation of the mock sample

Communities of small invertebrates were collected using Autonomous Reef Monitoring Structures (ARMS) in the Indian River Lagoon of Florida following a standard protocol described previously (*Leray & Knowlton, 2015*). Specimens larger than 2 mm were anesthetized, photographed alive to document color patterns and preserved in 95% EtOH. A total of 34 specimens, each belonging to a different morphological species in the phyla Arthropoda, Annelida, Echinodermata, Platyhelminthes, Chordata and Mollusca, were selected to make up the mock sample (Fig. 1). A small piece of tissue was subsampled from each specimen (Arthropoda: a leg or a section of a leg; Annelida and Platyhelminthes: a small section of the body; Chordata: muscle tissue; Echinodermata: a portion of the underside of the disk; Mollusca: a piece of the mantle) and placed in Costar plates (Corning) for

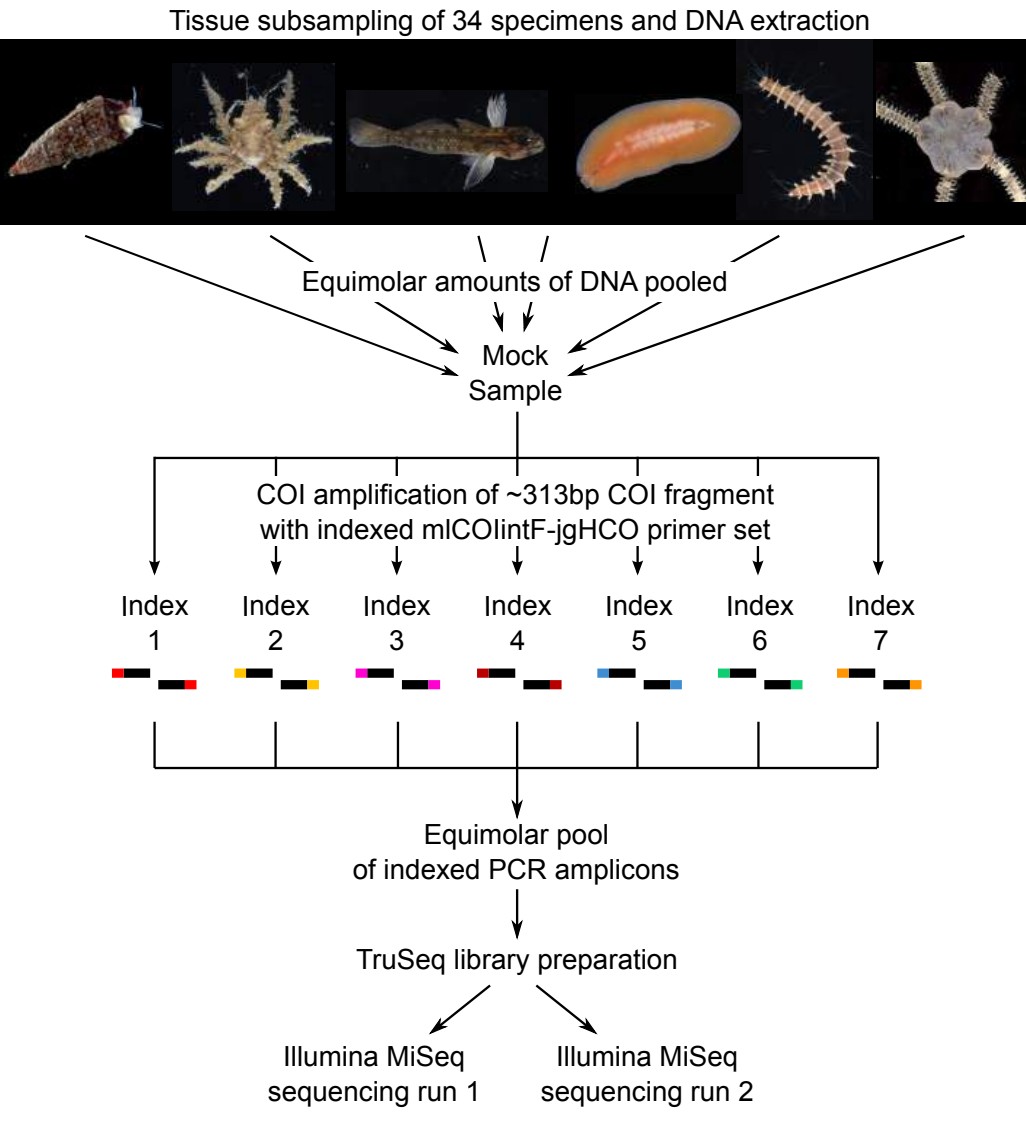

**Figure 1** **Diagram illustrating the study design.** Photo credit: Matthieu Leray.

phenol chloroform DNA extraction using the AutoGenprep 965 (Autogen). A negative control extraction was performed to check for potential contaminants. Eluted DNA of each individual extract was quantified with a Qubit fluorometer (dsDNA HS Assay kit; Invitrogen), diluted to approximately 5 ng/µl, and equimolar amounts of DNA pooled to make up the mock sample. Tissue subsampling and DNA extractions were conducted in a room free of PCR amplicons at the Laboratory of Analytical Biology of the Smithsonian National Museum of Natural History (Washington, D.C., USA).

## Reference barcode library

Individual DNA extracts were used for PCR amplification and Sanger sequencing of a ~658 bp portion of the COI gene as described previously (*Leray & Knowlton, 2015*). Sequences obtained in both directions were assembled, checked for stop codons and

frameshifts, and aligned in Geneious (Biomatters) to build a reference library containing COI barcodes for each of the 34 specimens included in the mock sample. The complete list of specimens, including taxonomy and GenBank accession numbers, are provided in Table S1. A PCR reaction performed with the negative control extraction confirmed the absence of contaminants (no band on 1.5% agarose gel).

## Illumina metabarcoding of the mock sample

Pooled genomic DNA (the mock sample) was used to amplify a ∼313 bp of the COI region with seven sets of indexed PCR primers (index-mlCOIintF/jgHCO-index (*Leray et al., 2013*)). The same index was used on the forward and reverse primer within each individual PCR amplification (e.g., Primer set 1: Index1-mlCOIF/jgHCO-index1; Primer set 2: Index2-mlCOIF/jgHCO-index2) (Fig. 1, Table 1) to minimize the likelihood of false read-to-sample assignments due to tag jumping (*Schnell, Bohmann & Gilbert, 2015*). Three PCR reactions (i.e., triplicates) were performed using each of the seven indexed primer pairs for a total of 21 PCRs using the PCR mixtures and touchdown temperature profile described previously (*Leray & Knowlton, 2015*). PCR reactions were prepared in a room free of PCR amplicons. Triplicate PCRs were pooled (for a total of seven pools) to limit stochastic amplification biases, and purified using Agencourt AMPure XP beads. Cleaned PCR products were quantified with a Qubit fluorometer, and equimolar amounts were combined into a single tube prior to library preparation using an adaptor ligation approach (*Leray, Haenel & Bourlat, 2016*). A total of 1 µg of pooled amplicon in a final volume of 60 µl was used for end repair, A-tailing and adaptor ligation following protocols and reagents of the TruSeq PCR-free kit (Illumina). The addition of Illumina indexed-adaptors via ligation prevents tag jumping caused by the formation of chimeric sequences during the bulk amplification of tagged amplicons (cf. dual-PCR approach, *Bourlat et al., 2016*). The library was validated via qPCR using the KAPA library quantification kit and diluted to a final concentration of 4nM. Paired-end sequencing of the library was performed twice on two separate runs of the Illumina MiSeq platform at the Laboratory of Analytical Biology of the Smithsonian National Museum of Natural History (Washington, D.C., USA) using the Miseq reagent kit v3 (600 cycles). The same MiSeq instrument was used for both runs conducted one week apart from each other. Note that the library of the present study was combined in each MiSeq run with several other unrelated libraries prepared with different TruSeq adaptors.

## Analysis of the Illumina Miseq data

The script used to analyse the dataset is provided in Text S1. Illumina forward and reverse reads were denoised using BFC (*Li, 2015*), an error correcting tool designed specifically for Illumina short reads. Denoised paired reads were merged into contigs in Usearch 8.1 (*Edgar, 2010*) allowing for a maximum of five mismatches and a maximum of zero gaps in the overlapping region. Pairs were also discarded if the number of expected errors predicted by Phred scores after merging (fastq_merge_maxee) was higher than one (*Edgar & Flyvbjerg, 2015*). The FASTA file was then demultiplexed based on the primer index in Mothur (*Schloss et al., 2009*) and additional reads were discarded if they (1) had any

**Table 1  Indexed primers used in this study.**

| Primer label | Primer sequence (5′–3′) |
| --- | --- |
| mlCOIintF-Index1 | **AGACGC**GGWACWGGWTGAACWGTWTAYCCYCC |
| mlCOIintF-Index2 | **AGTGTA**GGWACWGGWTGAACWGTWTAYCCYCC |
| mlCOIintF-Index3 | **ACTAGC**GGWACWGGWTGAACWGTWTAYCCYCC |
| mlCOIintF-Index4 | **ACAGTC**GGWACWGGWTGAACWGTWTAYCCYCC |
| mlCOIintF-Index5 | **ATCGAC**GGWACWGGWTGAACWGTWTAYCCYCC |
| mlCOIintF-Index6 | **ATGTCG**GGWACWGGWTGAACWGTWTAYCCYCC |
| mlCOIintF-Index7 | **ATAGCA**GGWACWGGWTGAACWGTWTAYCCYCC |
| jgHCO-Index1 | **AGACGC**TAIACYTCIGGRTGICCRAARAAYCA |
| jgHCO-Index2 | **AGTGTA**TAIACYTCIGGRTGICCRAARAAYCA |
| jgHCO-Index3 | **ACTAGC**TAIACYTCIGGRTGICCRAARAAYCA |
| jgHCO-Index4 | **ACAGTC**TAIACYTCIGGRTGICCRAARAAYCA |
| jgHCO-Index5 | **ATCGAC**TAIACYTCIGGRTGICCRAARAAYCA |
| jgHCO-Index6 | **ATGTCG**TAIACYTCIGGRTGICCRAARAAYCA |
| jgHCO-Index7 | **ATAGCA**TAIACYTCIGGRTGICCRAARAAYCA |

mismatch in the primer and index region, (2) did not have the same index on the forward and reverse primer to prevent false assignments of reads due to tag jumping (*Schnell, Bohmann & Gilbert, 2015*), (3) had at least one homopolymer region longer than 8 bp, or (4) had any ambiguous base calls.

Sequences of both MiSeq runs were pooled, and the dataset was dereplicated in Mothur. The option "enrichAlignment" implemented in MACSE (*Ranwez et al., 2011*) was then used to align unique reads to the reference COI database of the Moorea Biocode project (7,675 sequences from 30 phyla represented) based on amino acid translations. We selected the invertebrate mitochondrial translation code to perform the alignment and detect interruptions in the open reading frame due to nucleotide substitution or nucleotide insertion/deletion. We only retained sequences without any stop codons or frameshifts for subsequent analysis. To further reduce the variability in the dataset, we used the preclustering approach implemented in Mothur to merge reads differing by three or fewer than three bases. This algorithm ranks sequences in order of their abundance and merges rare sequences with more abundant sequences within the threshold specified. Reads were then screened for chimeras using UCHIME (*Edgar et al., 2011*), and all remaining preclusters represented by a single sequence (singletons) were discarded.

High quality reads were clustered in OTUs using CROP (*Hao, Jiang & Chen, 2011*), a Bayesian model that delineates OTUs based on the natural distribution of sequence dissimilarity of the dataset. Rather than using a hard cut-off (e.g., 5%), CROP generates clusters within user-defined lower (-l) and upper (-u) bound levels of similarity to account for differences in rates of sequence evolution among taxonomic groups. Here, we defined -l 3 and -u 4 because it was shown to create OTUs that closely reflect morphological species grouping among marine invertebrates by providing the lowest frequency of false positives (splitting of taxa) and false negatives (lumping of taxa) (*Leray et al., 2013*).

CROP outputs a file with one representative sequence per OTU that we used for taxonomic assignments. First, we ran local BLASTn searches in Geneious (Biomatters) to compare each representative sequence to the reference database containing COI barcodes of the 34 specimens included in the mock sample. Second, we conducted BLASTn searches against the full GenBank database and used the BOLD (Barcode of Life Data systems) search engine (*Ratnasingham & Hebert, 2007*) against all barcode records to identify OTUs that did not match a specimen included in the mock sample. Third, whenever sequence similarity to a reference barcode was lower than 97%, we used a Bayesian phylogenetic approach implemented in the Statistical Assignment Package (SAP) to assign OTU representative sequences to a higher taxonomic group (see *Leray, Meyer & Mills, 2015* for further details).

Representative COI sequences (313 bp) were used to infer phylogenetic relationships between OTUs using a Maximum Likelihood approach. We selected the best tree of 1,000 maximum likelihood search replicates computed using the adaptive best tree search analyses implemented in GARLI v2.1 (*Zwickl, 2006*) through the GARLI web service (*Bazinet, Zwickl & Cummings, 2014*). We used a general time reversible nucleotide model with a proportion of invariant sites and among site rate heterogeneity modeled with a discrete gamma distribution (GTR + I + G). GARLI default settings were used, including stepwise-addition starting trees.

## Analysis of dissimilarity in OTU composition

Abundance data and taxonomic information for each OTU were summarized in a .biom formatted OTU table and imported into QIIME (*Caporaso et al., 2010*). To evaluate compositional dissimilarities among replicates (i.e., β diversity) the OTU table was used to calculate distance matrices based on the Jaccard and the Bray Curtis metrics in Qiime (beta_diversity.py with -m binary_jaccard, bray_curtis). Note that the term "pseudo- β" diversity may be more appropriate for describing dissimilarities in composition caused by methodological artifacts because they artificially inflate diversity (e.g., incomplete sampling, *Manter & Bakker, 2015*). However, we use the term β diversity throughout for simplification. Jaccard considers only the presence-absence of OTUs; a value of 0 indicates that samples have exactly the same OTU composition whereas a value of 1 indicates that samples do not have any OTUs in common. Because indexed primers and random sampling during sequencing are more likely to affect the repeatability of rare OTUs, we repeated calculations of pairwise distances after sequentially discarding rare OTUs from the OTU table. In brief, we repeated calculations of Jaccard after removing OTUs represented by less than three sequences from the dataset, then after removing OTUs represented by four sequences and so on, until a dataset from which all OTUs represented by less than 30 sequences were removed. On the other hand, Bray–Curtis takes into account differences in abundance of reads between samples; a value of 0 indicates that samples are exactly identical in terms of OTU composition and abundance of reads whereas a value of 1 indicates that samples do not have any OTUs in common. It is much less sensitive to rare OTUs, resulting in low distance values when communities have shared abundant OTUs.

We assessed bias by calculating the pairwise Jaccard and Bray–Curtis at two levels of replication: primer index replicate and sequencing replicate. Within-primers pairwise

dissimilarities were calculated between communities obtained using identical primer indices but sequenced in different MiSeq runs (i.e., primer1/run1 vs. primer1/run2; primer2/run1 vs. primer2/run2). Between-primers pairwise dissimilarities were calculated between communities obtained using different primer indices regardless of the sequencing run (i.e., primer1/run1 vs. primer2/run1; primer2/run1 vs. primer1/run2). Within-run pairwise dissimilarities were calculated between communities obtained within the same MiSeq sequencing run regardless of the indexed primer set used (i.e., primer1/run1 vs. primer2/run1; primer2/run1 vs. primer3/run1). Between-run pairwise dissimilarities were calculated between communities obtained in different MiSeq sequencing runs regardless of the indexed primer set used (i.e., primer1/run1 vs. primer1/run2; primer1/run1 vs. primer2/run2). Pairwise community dissimilarities calculated within any indexed primer resulted from discordance that occurred after library preparation. Dissimilarities calculated between and within MiSeq runs as well as between indexed primers within a run resulted from a combination of technical and random sampling artifacts occurring during PCR amplification, library preparation and sequencing (Table 2).

To further examine similarities in OTU composition, we calculated hierarchical cluster trees using an Unweighted Pair Group Method with Arithmetic mean (UPGMA) based on Jaccard and Bray–Curtis. Branch support was calculated by jackknifing the dataset 100 times using 75% of the sequences in the smallest sample (34,206). We also visualized Bray–Curtis differences between samples using a principal coordinate analyses (PCoA). The score of each OTU was plotted in 2-dimensional PCoA space to illustrate their influence on the dissimilarities between samples. Finally, we tested differences in OTU composition between primer indices and sequencing runs using permutational multivariate analysis (PERMANOVA, *Anderson, 2001*) computed using 10,000 permutations within the R package Vegan (*Oksanen et al., 2009*). Because differences in sequencing depth can affect estimates of α and β diversity, all analyses were repeated with a dataset rarefied down to the lowest number of sequences that a sample contained (45,609).

## RESULTS

### Summary of sequencing runs

Illumina MiSeq sequencing runs provided a total of 779,758 (run 1) and 745,490 (run 2) raw paired end reads, of which 580,938 (74.5%) and 562,507 (75.4%) were successfully merged into contigs. Most paired reads that failed to merge (95.5% and 95.7%, respectively) had a number of expected errors above one. A total of 80,035 and 94,590 additional reads were discarded because they had at least one mismatch in the primer or index region, had a different index on the forward and reverse primer, had at least one homopolymer region longer than 8bp, or had one or more ambiguous base calls. Out of the 500,903 (64.2%) and 467,917 (62.7%) remaining reads, 23,655 and 22,173 had at least one frameshift or stop codon. In total, we obtained a dataset with 469,352 (60.2%) and 438,752 (58.8%) high quality paired reads in Miseq run 1 and 2, respectively. Within each Miseq run, the number of reads per indexed PCR ranged from 50,150 to 75,916 (mean ± SD = 67,030 ± 12,010) and from 45,609 to 78,387 (mean ± SD = 62,664 ± 11,885),

**Table 2  Pairwise dissimilarities calculated among replicates (i.e., pseudo-$\beta$ diversity) to evaluate the effect of random sampling and technical artefacts.**

| Pairwise dissimilarity | Description of calculation | Interpretation of dissimilarities |
|---|---|---|
| Within indexed primers | Between communities obtained using identical primer indices but sequenced in different MiSeq run (i.e., primer1/run1 vs. primer1/run2; primer2/run1 vs. primer2/run2) | Result from random sampling during sequencing only |
| Between indexed primers | Between communities obtained using different primer indices regardless of the sequencing run (i.e., primer1/run1 vs. primer2/run1; primer2/run1 vs. primer1/run2) | |
| Within Illumina MiSeq run | Between communities obtained within the same MiSeq run regardless of the indexed primer set used (i.e., primer1/run1 vs. primer2/run1; primer2/run1 vs. primer3/run1) | Result from a combination of technical artefacts during PCR (e.g., amplification bias caused by primer indices) and random sampling during sequencing |
| Between Illumina MiSeq run | Between communities obtained in different MiSeq runs regardless of the indexed primer set used (i.e., primer1/run1 vs. primer1/run2; primer1/run1 vs. primer2/run2) | |

respectively. The raw Illumina MiSeq and the final dataset are available from Figshare (MiSeq Run1, R1 direction: https://dx.doi.org/10.6084/m9.figshare.4039821.v1; MiSeq Run1, R2 direction: https://dx.doi.org/10.6084/m9.figshare.4039860.v1; MiSeq Run2, R1 direction: https://dx.doi.org/10.6084/m9.figshare.4039893.v1; MiSeq Run2, R2 direction: https://dx.doi.org/10.6084/m9.figshare.4039899.v1).

## Diversity and abundance

The Bayesian clustering tool CROP delineated 128 OTUs. Six bacterial OTUs and two OTUs matching contaminants (*Homo sapiens* and a rodent) representing a total of 244 sequences were removed from the dataset, leaving a total of 120 eukaryotic OTUs. BLASTn searches against the reference barcode library revealed that 34 of these OTUs corresponded to the 34 species included in the mock sample. We will refer to these OTUs as "*target OTUs*". The 86 OTUs that did not match any species included in the mock sample are hereafter referred to as "*non-target OTUs*". Among them, 31 (25.8%) had >97% similarity to GenBank and BOLD sequences while 41 (34.2%) could be confidently assigned to higher taxonomic levels using the Bayesian phylogenetic approach implemented in SAP, leaving only 14 OTUs (11.7%) unidentified. As noted above, the 34 target OTUs belonged to six animal phyla (Table S1) whereas the 72 identified non-target OTUs were more diverse, with representatives of 12 phyla (Fig. 2, Fig. S1). Among them, Arthropoda (23 OTUs), Annelida (11 OTUs), Rhodophyta (8 OTUs) and Porifera (7 OTUs) were the most common. Other photosynthetic organisms included three diatoms (phylum Bacillariophyta) and two brown algae (phylum Ochrophyta).

Among non-target OTUs, 20 corresponded to animals that typically live attached to a hard substrate (e.g., Hydrozoa, Ascidiacea). Moreover, one OTU belonged to a group of parasitic copepods known to live on fish hosts (Siphonostomatoida). A total of 44

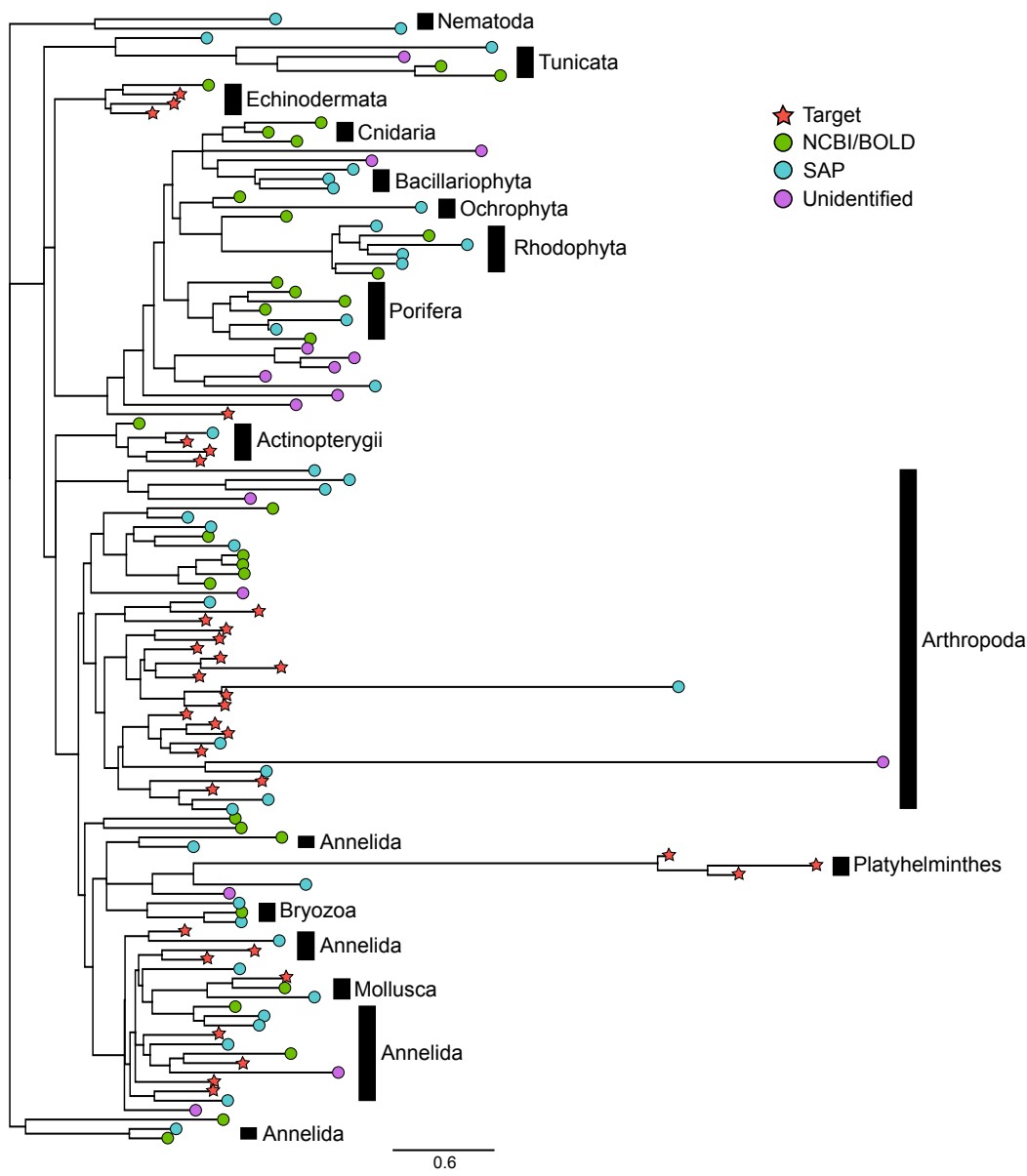

**Figure 2** **Phylogenetic relationships between representative COI sequences (313 bp) of 120 OTUs detected in the mock sample.** This is the best tree of 1,000 maximum likelihood search replicates computed using the adaptive best tree search analyses implemented in GARLI v2.1 (*Zwickl, 2006*) through the GARLI web service (*Bazinet, Zwickl & Cummings, 2014*) . The full tree is provided in Fig. S1. Branch tip symbols indicate the method of identification of each OTU. Target: OTU that matched the COI sequence of a species included in the mock sample (referred to as "target *OTUs*" throughout the manuscript; note that OTUs that did not match any target OTU are referred to as "*non-target OTUs*" throughout the manuscript); NCBI/BOLD, OTU that did not match a target OTU but had >97% similarity with a reference COI barcode in NCBI or BOLD; SAP, OTU that did not match a target OTU nor a reference COI barcode in NCBI or BOLD but could be confidently assigned to higher a taxonomic level using a Bayesian phylogenetic approach implemented in the Statistical Assignment Package (SAP); Unidentified, OTU that could not be confidently identified to any taxonomic group using the three approaches detailed above.

additional OTUs (51%) matched species or OTUs previously reported in the Indian River Lagoon of Florida ((*Leray & Knowlton, 2015*), http://www.sms.si.edu/IRLspec/).

The rank abundance curve showed relatively few common OTUs followed by a long tail of rare taxa (Fig. 3). The 34 target OTUs were by far the most represented, accounting for 99.7% of the reads in each of two Miseq runs. In contrast, the 86 non-target OTUs accounted for only 0.3% of reads (1,351 and 1,289 reads in each run). Only one of these had more sequence reads than some of the target OTUs; it accounted for 1,162 and 1,018 reads in Miseq run 1 and 2, respectively, but was unidentified. Most of the non-target OTUs were very rare in the dataset (79 of 86 with <10 sequences).

Despite pooling equimolar amounts of total genomic DNA, the total number of reads for each target OTU spanned three orders of magnitude. Based on expected number of reads calculated using the total amount of genomic DNA, 8 OTUs were highly under-represented (<4,000 sequences), while two taxa were highly over-represented (>80,000 sequences) (Fig. 4A). At the phylum level, the bias was less pronounced, but observed proportions of reads also differed from what was expected based on ratios of total genomic DNA pooled (Fig. 4B). Arthropoda (rank 1, Fig. 4B) and Platyhelminthes (rank 6, Fig. 4B) were underrepresented while Annelida (rank 2, Fig. 4B) was over-represented.

The 34 target OTUs were present in every one of the seven different indexed PCR trials (mean ± SD = 34 ± 0.0). On the other hand, the 86 non-target OTUs appeared much less consistently; the mean (±SD) number of non-target OTUs per indexed PCR trial was 16 (±4.4) and 21 (±3.7) in MiSeq runs 1 and 2, respectively. After rarefying the dataset down to the lowest number of reads (45,609) to account for differences in sequencing depth, all target OTUs were still detected in all seven indexed PCR trials, whereas the mean (±SD) number of non-target OTUs per indexed PCR trial was only 12.6 (±3.9) and 18.6 (±2.9) in the two sequencing runs. The mean total number of OTUs was significantly different between MiSeq runs based on both the non-rarefied ($t = -2.28$, $df = 11.6$, $P = 0.04$) and the rarefied ($t = -3.27$, $df = 11.2$, $P = 0.01$) datasets.

## Dissimilarity in OTU composition

A total of 40 of the 86 non-target OTUs (46%) were only present in one of the two MiSeq runs, a majority of which (26) were represented by only two sequences. Rarefying the OTU table yielded similar results (e.g., 44 OTUs were present in a single MiSeq run). We repeated the rarefaction draws five times and found no differences in the general pattern as a result of random subsampling (Fig. S2). We present the results obtained using one of the rarefied datasets (draw 1 in Fig. S2) below.

Pairwise β Jaccard dissimilarities (compositional similarities among replicates in terms of presence-absence) ranged from 0.24 to 0.43 between runs and from 0.18 to 0.45 within a run (see Table 2 for a summary of pairwise dissimilarity calculations). Values of β Jaccard also ranged from 0.28 and 0.39 within indexed primers across sequencing runs because of the presence of rare OTUs.

There was a relationship between the presence of non-target OTUs in indexed PCR trials and the number of reads they represented in the dataset (Fig. 5). As a result, values of β Jaccard dissimilarities gradually decreased with the removal of rare OTUs (Fig. 6). Pairwise

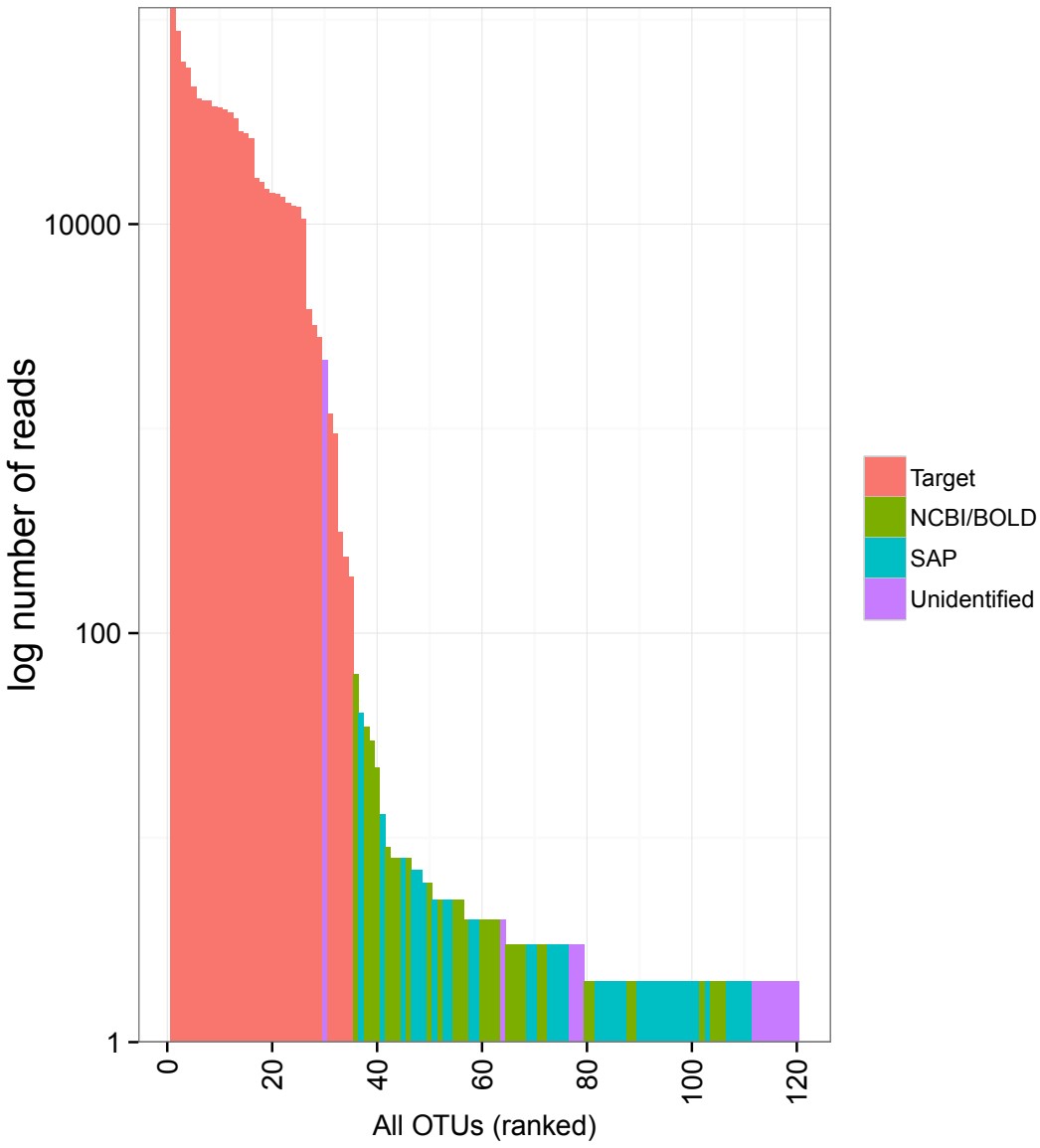

**Figure 3** **Rank abundance plot of OTUs detected in the mock sample.** Colours indicate the method of identification of each OTU (see Fig. 2 legend). Target: OTU that matched the COI sequence of a species included in the mock sample (referred to as ''target *OTUs*'' throughout the manuscript); NCBI/BOLD, OTU that did not match a target OTU but had >97% similarity with a reference COI barcode in NCBI or BOLD; SAP, OTU that did not match a target OTU nor a reference COI barcode in NCBI or BOLD but could be confidently assigned to higher a taxonomic level using a Bayesian phylogenetic approach implemented in the Statistical Assignment Package (SAP); Unidentified, OTU that could not be confidently identified to any taxonomic group using the three approaches detailed above.

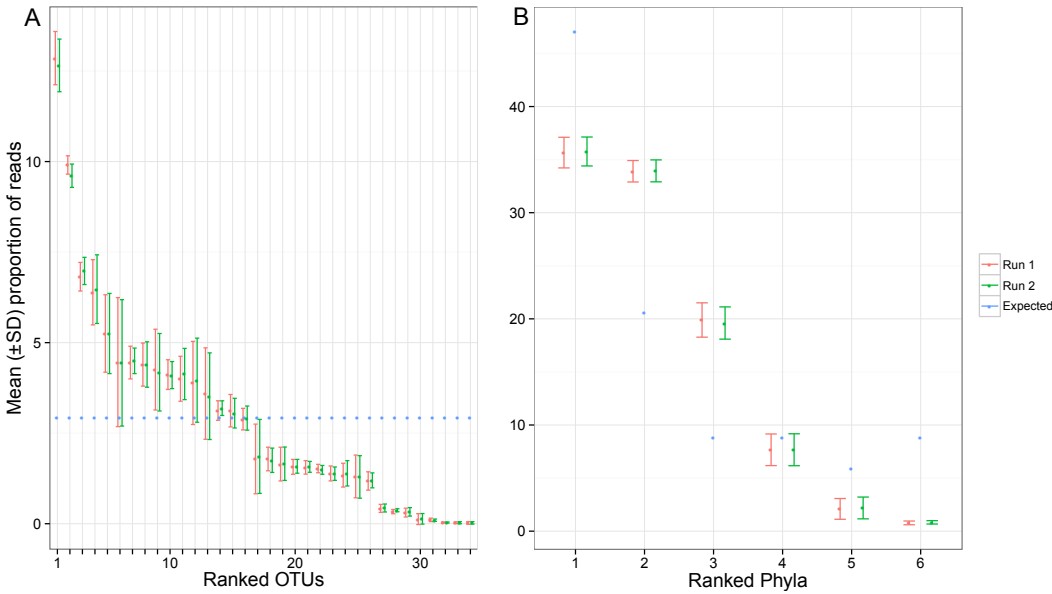

**Figure 4** **Proportion of reads recovered for each of the target OTUs separately (A) and after combining them per phyla (B).** Mean and standard deviation (SD) were calculated among seven indexed PCRs. The expected proportion of reads was calculated based on the proportion of total genomic DNA pooled (nuclear and mitochondrial DNA). Ranked phyla are as follows: (1) Arthorpoda; (2) Annelida; (3) Chordata; (4) Echinodermata; (5) Mollusca; (6) Platyhelminthes. The identity of the ranked OTUs can be found in Table S1.

values calculated within and between runs were close to 0 after all OTUs represented by less than eight sequences were removed (Figs. 6A and 6B). The same pattern was observed within and between indexed primers (Figs. 6C and 6D). All values of β Jaccard dissimilarities were equal to 0 (composition exactly the same) after OTUs with less than 30 sequences were removed from the dataset.

β Bray–Curtis dissimilarity values ranged from 0.01 to 0.14 between runs and from 0.04 to 0.14 within a run (Fig. 7). Values of β Bray–Curtis also ranged between 0.04 and 0.14 between the seven indexed primer trials, but they remained between 0.01 and 0.02 within indexed primers.

UPGMA trees for the seven indexed PCR trials and two sequencing runs (14 total communities) based on the Jaccard (presence-absence) analysis showed that there was no significant grouping among indexed PCR trials or among Illumina runs (Fig. 8A). Indexed PCRs explained a higher fraction of the variance in Jaccard dissimilarity ($R^2 = 47.4\%$) than Illumina runs ($R^2 = 7.8\%$) in PERMANOVA analysis (Table 3), but there were no significant differences between indexed PCRs ($F_{(1,6)} = 1.059$, $p = 0.367$) or between Illumina runs ($F_{(1,6)} = 1.049$, $p = 0.414$). Calculations based on an OTU table from which the rare OTUs (<8 sequences) were removed showed a similar lack of structure (Fig. 8B). On the other hand, the Bray–Curtis UPGMA tree displayed a consistent clustering of indexed PCR trials obtained from the two Ilumina MiSeq runs (Fig. 8C). PERMANOVA analysis confirmed that much of the variation in read abundance was driven by indexed PCRs ($R^2 = 98.3\%$, $F_{(1,6)} = 106.886$, $p < 0.001$) but differences were also significant

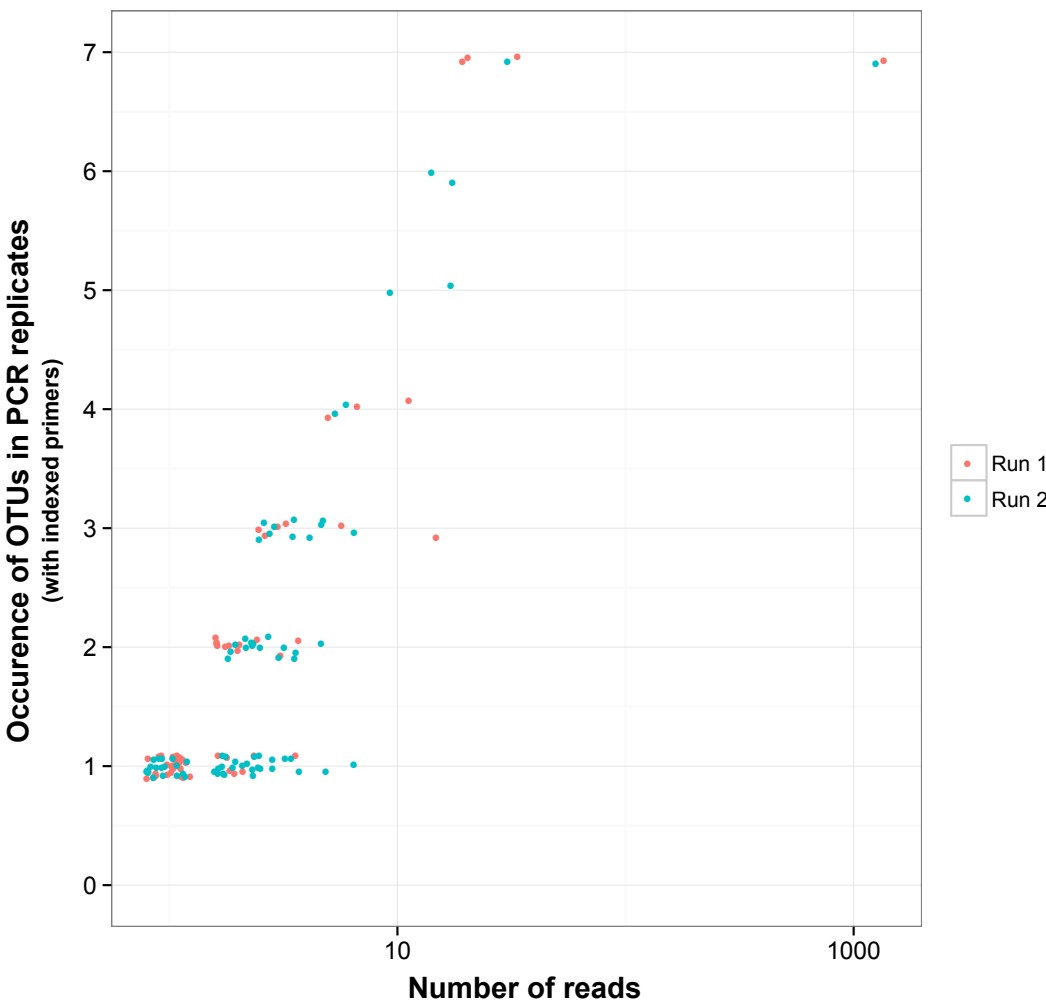

**Figure 5** **Reproducibility of non-target OTUs in indexed PCRs as a function of the total number of reads they represent in each sequencing run.** "*Non-target OTUs*" correspond to OTUs that did not match the COI sequence of a species included in the mock sample. Each point represents data for one sequencing run for one non-target OTU, so that for example, four non-target OTUs found in all seven indexed PCR trials in sequencing run 1 and two non-target OTUs found in all seven PCR trials in sequencing run 2 had a total number of reads >10 for both sequencing runs.

between Illumina runs ($R^2 = 0.7\%$, $F_{(1,6)} = 4.888$, $p < 0.001$). Indexed primer sets 2 and 3 appeared as outliers on the UPGMA tree (Fig. 8C) and PCoA plot (Fig. 9). OTUs 11 (*Cilicaea*) and 34 (Polychaeta) drove differences between primer set 2 and the rest of the PCR trials while OTUs 80 (*Hypleurochilus geminatus*), 123 (*Menippe mercenaria*) and 114 (Polychaeta) played a significant role at differentiating primer set 3 from the rest of the dataset.

## DISCUSSION

Before metabarcoding can be routine in biodiversity monitoring, it is essential to quantify all potential sources of variability in diversity estimates. Our study aimed to evaluate the

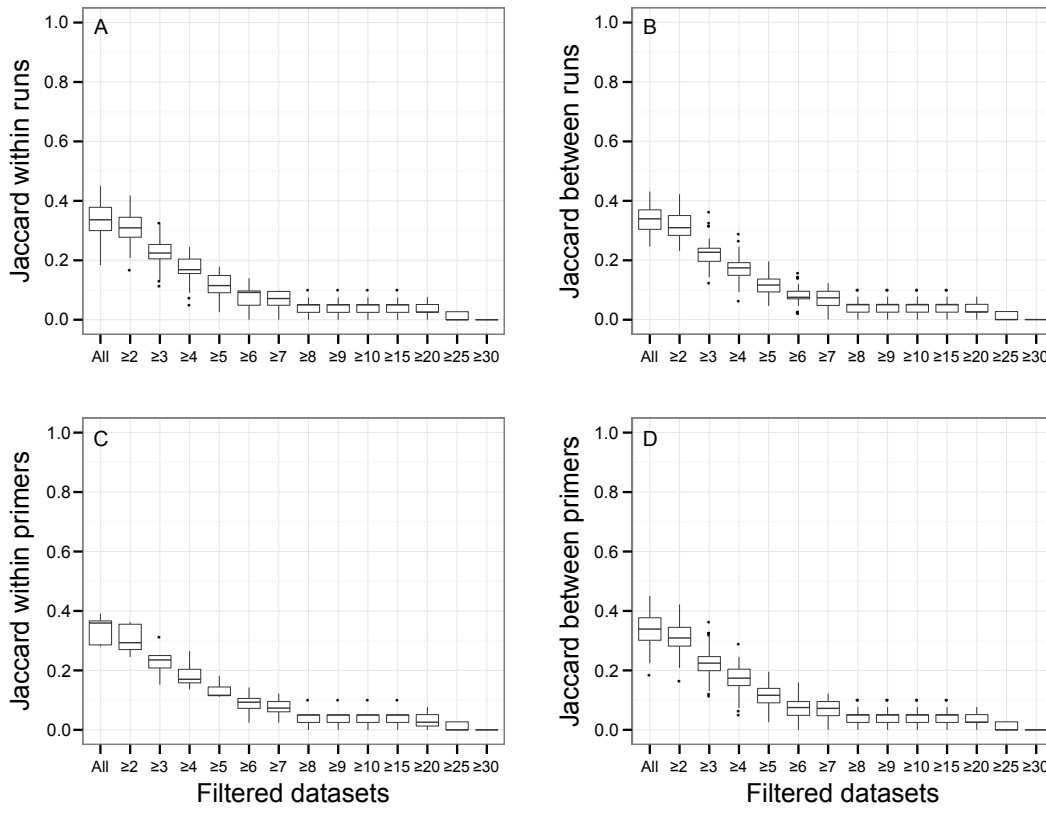

**Figure 6  Jaccard pairwise dissimilarity among indexed PCRs and Illumina Miseq runs.** The effect of rare OTUs on Jaccard was evaluated by sequentially removing low abundance OTUs from the rarefied OTU table (all: full rarefied OTU table; ≥2: rarefied OTU table without singletons; ≥3: rarefied OTU table without singletons and doubletons). Jaccard value of 0 indicates that samples have exactly the same OTU composition whereas a value of 1 indicates that samples do have any OTU in common.

**Table 3  Permutational multivariate analysis (PERMANOVA) testing differences in OTU composition between indexed PCRs and Illumina MiSeq runs.** Jaccard considers only the presence-absence of OTUs whereas Bray–Curtis also takes into account differences in abundance of reads between samples.

|  | Source of variation | *Df* | *SS* | *F Model* | *R²* | *P-value* |
|---|---|---|---|---|---|---|
| Jaccard | MiSeq Runs | 1 | 0.059 | 1.049 | 0.078 | 0.414 |
|  | Indexed PCRs | 6 | 0.357 | 1.059 | 0.474 | 0.367 |
|  | Residuals | 6 | 0.337 |  | 0.448 |  |
| Bray–Curtis | MiSeq Runs | 1 | 0.000 | 4.888 | 0.007 | <0.001 |
|  | Indexed PCRs | 6 | 0.051 | 106.886 | 0.983 | <0.001 |
|  | Residuals | 6 | 0.000 |  | 0.009 |  |

**Notes.**
*Df*, degrees of freedom; *SS*, sum of squares; *F Model*, *F* value by permutation.
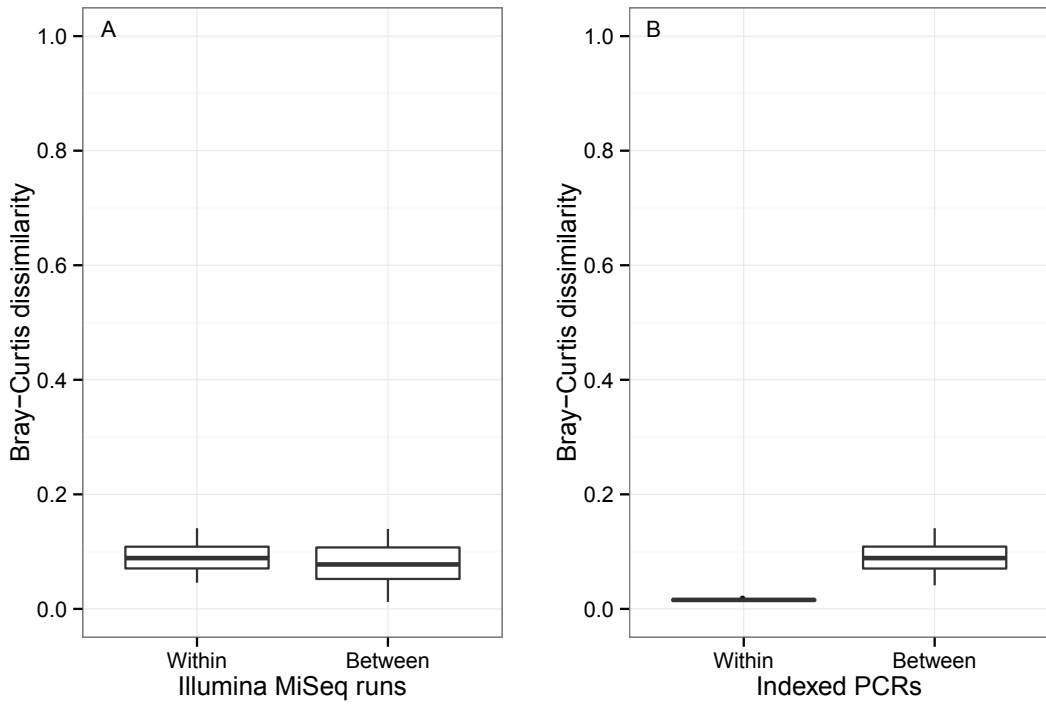

**Figure 7** **Bray–Curtis pairwise dissimilarity among indexed PCRs and Illumina Miseq runs.** Bray–Curtis value of 0 indicates that samples are identical in terms of OTU composition and abundance of reads whereas a value of 1 indicates that samples do not have any OTUs in common.

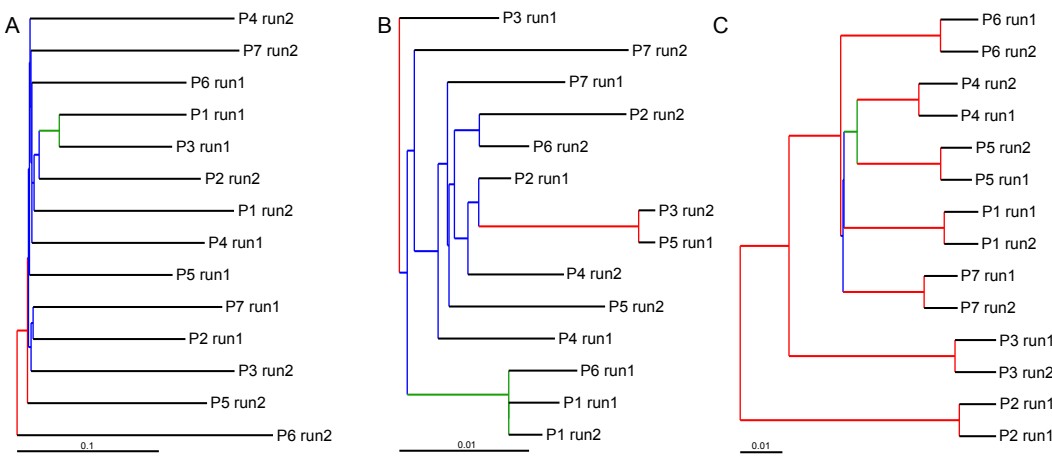

**Figure 8** **Hierarchical cluster trees constructed using UPGMA with jackknife support to depict similarities in OTU composition between indexed PCRs and Illumina MiSeq runs.** Similarities between samples were calculated using the incidence-based Jaccard (A and B) and abundance-based Bray–Curtis (C) indices. All OTUs comprising less than eight sequences were considered rare in (B). Branch colors illustrate the level of jackknife support. Red: 75–100%; Orange: 50–75%; Green: 25–50%; Blue: 0–25%. P, Indexed primer pair number.

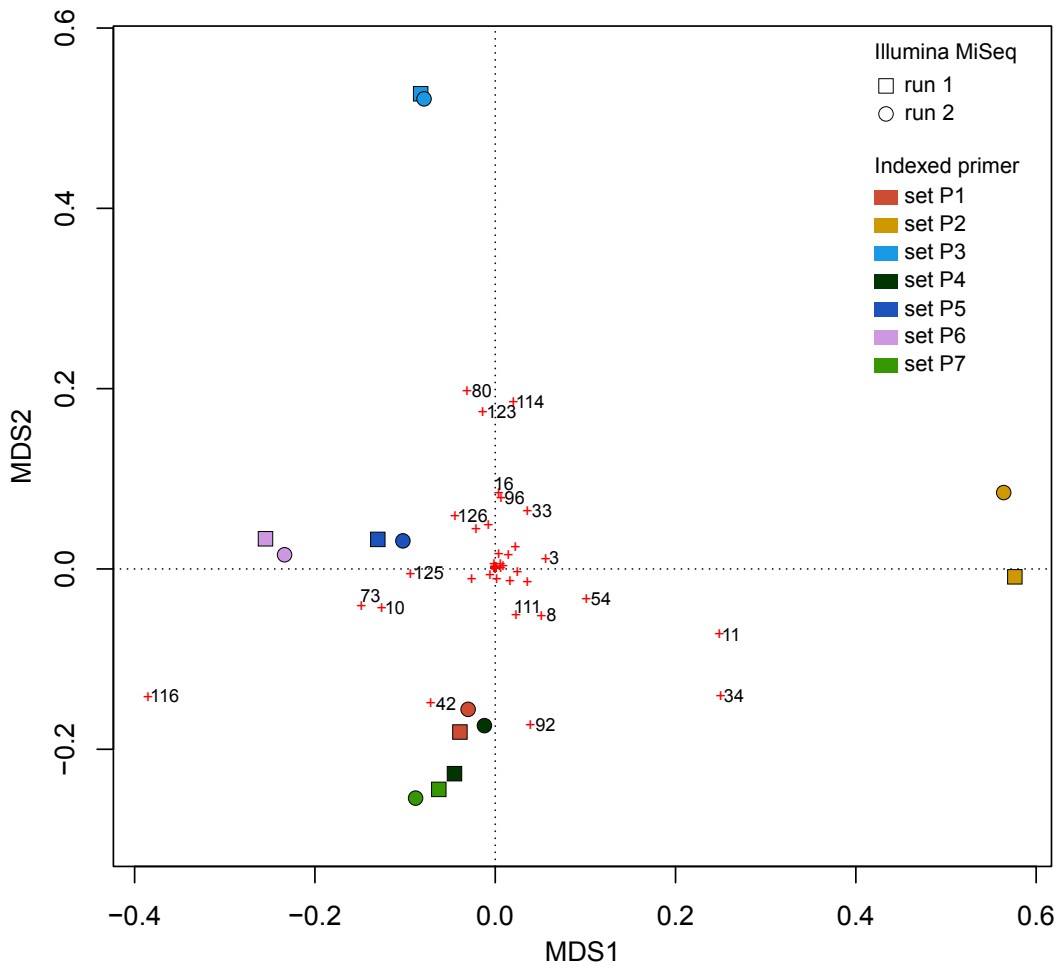

**Figure 9** **Principal Coordinate Analyses (PCoA) depicting Bray–Curtis dissimilarities between samples.** Red crosses represent the score of each OTU in 2-dimensional PCoA. Refer to the full OTU table in Table S2 to link OTU numbers to taxonomic names.

effect of indexed PCRs and random sampling in MiSeq sequencing on the reproducibility of metabarcoding.

We recovered a much higher total number of OTUs (120) than expected based on the known composition of the mock sample of 34 organisms. However, most of these non-target sequences could be identified, and we used stringent data acceptance protocols, suggesting that most of these sequences represented genuine but rare OTUs (i.e., taxa or products of taxa that were present in trace amounts in the original mock sample but undetected when it was created) rather than DNA artifacts. Most non-target OTUs were represented by very few sequences (with one notable exception), but the relative abundance of target OTUs varied by up to three orders of magnitude.

By comparing results between different index PCR trials and sequencing runs, we show that the noise in presence–absence data between replicates is most likely the result of random sampling of rare taxa, which causes the low reproducibility of rare OTUs. On the

other hand, we found that indexed PCRs accounted for a larger amount of variation in read numbers than random sampling during sequencing.

Metabarcoding is known to be a much more sensitive measure of biodiversity than traditional morphological surveys because any trace of DNA in the sample is potentially detectable. As such, the presence of numerous OTUs (86) that did not match any of the 34 species comprising the mock sample is not surprising. Species level assignment of 31 OTUs to GenBank and BOLD reference barcodes (>97% similarity) confirms that they all belonged to the local Floridian fauna, not common laboratory contaminants. A total of 20 sessile and 13 protist taxa were likely present on the legs of spider crabs and other arthropods. In fact, epiphytes, endosymbionts and also parasites are commonly described in sequence libraries (*Geisen et al., 2015*; *de Vargas et al., 2015*). Motile OTUs (e.g., among arthropods, molluscs and echinoderms) may be part of the diet of carnivorous annelids and thus present in the section of their digestive tract that was used for DNA extraction. Sequences of non-target OTUs may also originate from eDNA traces (*Kelly et al., 2014*), such as excreted cells, faeces, epidermal tissue, hair and body fluids from dead individuals, especially given that many specimens were in close proximity in a tray for several hours during field collecting. Interestingly, 14 OTUs could not confidently be assigned to any taxonomic level using the Bayesian Phylogenetic approach implemented in SAP. According to their phylogenetic placement (Fig. 2, Fig. S1) and their most similar BLASTn hits in GenBank and BOLD (not shown), most of these OTUs likely belong to single-celled eukaryotes that are highly underrepresented in public databases.

Alternatively, the presence of artifacts among unidentified non-target OTUs cannot be excluded despite stringent molecular (e.g., negative DNA extractions and PCRs, separate pre- and post-PCR areas) and data analysis (e.g., amino acid alignment) procedures. For example, spurious OTUs that result from sequencing errors or PCR chimeras may still account for some of the non-target OTUs (e.g., long branches, Fig. 2), but they most likely represent a minor fraction of the reads. Non-functional copies of COI transposed to the nuclear genome (numts) may also inflate diversity if they have not accumulated enough variation for interruptions in the open reading frame to occur. Numts are relatively common among marine invertebrates (e.g., decapods, *Williams & Knowlton, 2001*) but a majority include stop codons and frameshifts (*Song et al., 2008*) and have therefore been discarded. PCR and sequencing of COI transcripts obtained from the reverse transcription of cDNA from mRNA would ultimately ensure the removal of all non-functional copies (*Williams & Knowlton, 2001*; *Machida et al., 2009*). Contaminations across libraries during the sequencing reaction are another potential source of foreign DNA. This occurs when mixed clusters form on the Illumina flowcell during bridge amplification. This phenomenon may result in reads that have a different Illumina index than the one they were assigned (cf. bleeding effect; *Kircher, Sawyer & Meyer, 2012*; *Schnell, Bohmann & Gilbert, 2015*). We used local BLASTn searches to compare OTUs detected in the present study to sequences from libraries (independent studies) sequenced simultaneously in the MiSeq runs. We found that only eight rare OTUs had a >97% similarity between the two datasets which confirms previous findings (*Schnell, Bohmann & Gilbert, 2015*) that cross-library contamination is not a major source of error.

Our results confirm the existence of an amplification bias between taxa. While most target OTUs were represented by between ~10,000 and ~60,000 sequences, a few were either highly over- or under-represented. As observed previously (*Leray & Knowlton, 2015*), effects of primer biases tend to decrease as we group OTUs in higher taxonomic levels (Fig. 4B) which suggests that the number of reads could be used as proxy for biomass of functional groups. Importantly, some biological factors may also affect quantitative estimates. For example, amounts of mitochondrial DNA differ between and even within tissue types as a function of energetic demand (*Fernandez-Vizarra et al., 2011*). Therefore, our subsampling strategy may also account for some of the variation observed.

Despite its increasing use for biodiversity monitoring, the reproducibility of metabarcoding datasets has only rarely been evaluated. Our study demonstrates that the reproducibility of OTUs can be predicted from the total number of reads they represent in the dataset. Abundant OTUs are highly reproducible, whereas low abundance (rare) OTUs were never consistently found across PCR replicates and sequencing runs. These findings have important implications for estimates of β diversity calculated based on presence-absence data because the inclusion of rare OTUs may induce a background noise (high β diversity) that could mask real ecological patterns.

The high level of noise between the two replicate sequencing runs is one of the most surprising findings of this study. Because sequencing replicates were derived from an individual library, the discrepancy in presence-absence data must have occurred during the generation of the templates on the flow cell. McIntyre and colleagues (*2011*) reported similar inconsistencies in the detection of low-abundance transcripts and proposed that the observed level of noise was expected under a scenario of random variation. They calculated that less than 0.002% of the total given molecules loaded on an Illumina platform were effectively sequenced because a very small portion binds to the flow cell.

Our initial assumption was that randomness would account for a minor level of variation in OTU presence-absence in comparison to the noise caused by indexed PCR primers. This is because previous studies have shown that small tags could affect the binding efficiency of primers across taxa (*O'Donnell et al., 2016*). However, our results do not support this. First, the effect of rare OTUs was nearly as pronounced on β Jaccard values within indexed primer trials (Fig. 6C) (random sampling only), as it was between indexed primers as well as within and between sequencing runs (Figs. 6D, 6A and 6B respectively) (random sampling combined with potential technical artifacts). Second, despite accounting for the largest fraction of the variance in Jaccard dissimilarity (47.4%, Table 3), differences between indexed PCRs were not significant in PERMANOVA analysis, and indexed primers did not cluster together on UPGMA trees (e.g., Indexed P1 from run 1 and 2 are not grouped together (Figs. 8A and 8B )). This indicates that levels of technical biases due to primer indices are not a significant factor affecting the presence-absence of OTUs. On the other hand, our results show that indexed PCRs induced a higher level of noise in relative abundance data than sequencing alone (Fig. 8C, Table 3). The lack of evidence for taxon-specific biases suggests that a combination of technical factors (e.g., minor differences in binding efficiency) and stochastic events during early PCR cycles could be responsible for this pattern. A double PCR approach, that consists of a first PCR of genomic

DNA template with non-indexed primers, followed by a second PCR of the template of the first reaction with indexed primer (*Bourlat et al., 2016*), might help alleviate differences among replicates if the noise is caused by the indices (*Berry et al., 2011*; *O'Donnell et al., 2016*). Ultimately, bioinformatics advances should make it possible to use non-PCR based approaches on complex environmental samples of eukaryotes (*Tang et al., 2015*).

## CONCLUSION

Our study demonstrates that the random sampling effect of the sequencing process causes differences among replicates, artificially inflating β-diversity. Its effect is particularly significant on presence-absence data that is deemed the most reliable product of metabarcoding experiments (as opposed to abundance data). On the other hand, we show that many of the less reproducible OTUs represent genuine taxa occurring in low abundance in the community. Therefore, the strategy for handling rare OTUs should vary according to the objectives of the study and how sensitive the interpretation of the data may be to the presence of false positives or false negatives.

In environmental biomonitoring studies that use common β descriptors based on presence-absence, non-reproducible OTUs may obscure patterns of community turn-over through time or along spatial gradients. As a result, some authors have recommended the systematic removal of rare OTUs such as doubletons and tripletons (*Kunin et al., 2010*). Here, our dataset shows that the cut-off for reproducibility across all replicates is much higher than previously suggested. β Jaccard diversity reached values close to 0 only if OTUs containing less than eight sequences (0.02% of reads) were removed, which comes at the price of discarding valuable ecological data (e.g., true detection of epiphytes). However, note that alternative β diversity metrics that give less weight to rare taxa will be less sensitive to undersampling (*Beck, Holloway & Schwanghart, 2013*) and consequently, allow more relaxed cut-offs.

The systematic exclusion of rare OTUs, under the assumption that they are most likely false positive records, will be particularly detrimental to studies seeking to investigate the distribution of specific taxa. For example, non-indigenous species present in the form of propagules in the environment may not be consistently detected across replicates because of their low relative abundance (*Zaiko et al., 2015*). Similarly, such an approach may underestimate the contribution of highly digested and small prey items in food webs (*Leray, Meyer & Mills, 2015*). Rather than systematically removing OTUs based on arbitrary detection cut-offs, some have therefore advocated coupling metabarcoding data with statistical methods that account for the presence of false positives (*Lahoz-Monfort, Guillera-Arroita & Tingley, 2016*). Given replicate PCR assays or replicate samples within sites, occupancy models infer the probability of detection of individual species given the rate of imperfect detection (i.e., false positives). For example, this approach was successfully used to estimate the probability of occurrence of the invasive Burmese python in South Florida (*Hunter et al., 2015*) with important conservation implications. When possible, the simultaneous collection of data less prone to false positives (e.g., visual surveys) may prove useful to corroborate the presence of ambiguous detections (i.e., taxa that are not detected

across all PCR replicates) and estimate the probability of false negatives in site occupancy models (*Chambert, Miller & Nichols, 2015*; *Miller et al., 2015*).

Regardless of the application, our results further reinforce the importance of the inclusion of technical replicates (parallel PCR and sequencing from the same sample) in metabarcoding experimental designs. Data reproducibility should be determined empirically as it will depend upon the sequencing depth, the type of sample, the sequence analysis pipeline, and the number of replicates.

## ACKNOWLEDGEMENTS

We thank Amanda Devine and Jeffrey Hunt for logistical support as well as Matthew Kweskin and Paul Frandsen for informatics support. James Lawrence O'Donnell and Emmett Duffy provided constructive comments on an earlier version of the manuscript. We are grateful to Ryan Kelly and Rupert Collins for their helpful suggestions that improved the clarity of the manuscript as well as Tomas Hrbek for editing the paper. This is contribution #13 from the Smithsonian's MarineGEO Network and contribution #1049 from the Smithsonian Marine Station at Fort Pierce.

### Funding

Financial support was provided by the Sant Chair and Smithsonian Tennenbaum Marine Observatories Network. All of the laboratory and data analysis were conducted in and with the support of the L.A.B. facilities of the Smithsonian National Museum of Natural History. The funders had no role in study design, data collection and analysis, decision to publish, or preparation of the manuscript.

### Grant Disclosures

The following grant information was disclosed by the authors:
Sant Chair and Smithsonian Tennenbaum Marine Observatories Network.
Smithsonian National Museum of Natural History.

### Competing Interests

The authors declare there are no competing interests.

### Author Contributions

- Matthieu Leray conceived and designed the experiments, performed the experiments, analyzed the data, wrote the paper, prepared figures and/or tables, reviewed drafts of the paper.
- Nancy Knowlton conceived and designed the experiments, contributed reagents/materials/analysis tools, wrote the paper, reviewed drafts of the paper.

### Field Study Permissions

The following information was supplied relating to field study approvals (i.e., approving body and any reference numbers):

Research Permit SAJ-2012-02893(NW-SLR) was provided by the US Army Corps of Engineers to sample in the Indian River Lagoon in Florida.

### DNA Deposition

The following information was supplied regarding the deposition of DNA sequences:

The complete list of specimens, including taxonomy and GenBank accession numbers, are provided in Table S1.

The raw Illumina MiSeq and the final dataset are available from Figshare (MiSeq Run1, R1 direction: https://dx.doi.org/10.6084/m9.figshare.4039821.v1; MiSeq Run1, R2 direction: https://dx.doi.org/10.6084/m9.figshare.4039860.v1; MiSeq Run2, R1 direction: https://dx.doi.org/10.6084/m9.figshare.4039893.v1; MiSeq Run2, R2 direction: https://dx.doi.org/10.6084/m9.figshare.4039899.v1).

### Supplemental Information

Supplemental information for this article can be found online at http://dx.doi.org/10.7717/peerj.3006#supplemental-information.

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
