# Peer review of "Random sampling causes the low reproducibility of rare eukaryotic OTUs in Illumina COI metabarcoding"

_PeerJ, doi:10.7717/peerj.3006_

## Round 0.1 · original submission · Minor Revisions

Dear Authors,

Thank you for submitting a fine MS to PeerJ. Both referees suggested minor revision, and I am also recommending minor revision.
In my opinion, the MS is well organized, informative and clear on the focal issues of this study. What I would like to see is discussion of, and dealing more efficiently with different types of rare OTUs. Basically things which could be lab contaminants, or contaminants from other samples that were run together with these samples. Other interesting thing to address is to look at the effects of pooling vs. not pooling runs on the different types of rare OTUs. In the revisions, I also think it would be highly useful to make use of the PERMANOVA framework, and this should help in differentiating the these different types of rare OTUs. And then based on these results, I suggest making good practice recommendations.

I look forward to seeing a revised version shortly.
Sincerely,

Tomas Hrbek

·

Basic reporting

No Comments

Experimental design

See comments below

Validity of the findings

See comments below

Additional comments

Leray and Knowlton report the results of a metabarcoding sequencing study in which they test for the sources of variability in the results of such studies. They find that: (1) all of the species of their mock community were detected, and that these constitute the vast majority of sequenced reads (although this was not the purpose of the study…it still seems an important and comforting result), (2) the rare tail of OTUs is stochastic, such that rare OTUs do not routinely co-occur in replicate sequencing samples. Because most OTUs in metabarcoding studies tend to be very rare, this means that overall reproducibility is likely to be low if one cares about these rare OTUs, (3) the abundances of reads for species in the mock community were highly skewed, in part due to bias driven by PCR index sequences (although likely also due to amplification bias with the core set of PCR primers more generally, as the authors note). The take-home point (4) is that one should view these rare OTUs with skepticism and perhaps omit them during quality-control steps in metabarcoding studies. (I might suggest that a greater level of replication would also address the problem of discerning rare-true vs. rare-false OTUs, as I note below).

This is a useful and straightforward piece of work, and I look forward to seeing it in print and being able to cite it. I have a few comments, suggestions for strengthening, and clarifications that follow below, but on the whole I think this is nearly publishable as-is.

Big-picture points:
1. It would make sense to lay out all of the sources of variability that you can imagine, early in the paper, and then proceed by highlighting those sources that you are interrogating in this work. (Note, we attempted to do this in Shelton et al. 2016, http://onlinelibrary.wiley.com/doi/10.1890/15-1733.1/full, in the context of statistical modeling, but just a brief paragraph orienting the reader would do just fine in the present paper).
2. Consistent with this idea, I think it would simplify the analysis (and discussion of the analysis) to put the analysis of variance in an ANOVA-like framework. PERMANOVA (as implemented in vegan, in R) does this easily and well. You could then say that, e.g., “14% of the variance in community distance among samples was due to index sequences, 5% was due to sequencing runs, and the remainder due to stochastic PCR effects and other error.” Or something. You could then do this using two different metrics of community distance, as you have done throughout already (Jaccard and Bray-Curtis). Doing so would just put all of the results in one place (a table) that would be easy to read and understand.
3. Note that rarefaction adds an additional level of variability. Each rarefaction draw will differ from each other draw, and therefore so will your ultimate results. This will make little difference in Bray-Curtis but a bigger different in Jaccard. This is worth noting, and perhaps adding a supplemental figure illustrating the effects of different rarefaction on your results.
4. We have noticed that many of our rare, non-target reads (to use the language of this paper) come from some kind of cross-contamination within the same sequencing run. It would be very instructive to look at the other samples sequenced on the same MiSeq runs (Line 148; these were for other projects, presumably), and see if the taxa you are observing in this rare tail of OTUs is in fact derived from these other samples. To do this thoroughly, you would need to re-run your pipeline and match OTUs across projects. But even a cursory look at those other projects might tell you where some of this low-level signal is coming from.
5. If proposing to cut off the rare tail of OTUs for some purposes (e.g., Conclusion paragraph), it would be useful to address the pros/cons of the DeSeq2 approach (Love et al. 2014, https://genomebiology.biomedcentral.com/articles/10.1186/s13059-014-0550-8), as well as to address the possibility of non-arbitrary cutoffs provided for by a simple statistical model (e.g., Lahoz-Monfort et al 2015; http://onlinelibrary.wiley.com/doi/10.1111/1755-0998.12486/pdf).


More minor points:

1. Line 77 and throughout: I suggest using a different term other than “random sampling” (which is, of course, already a term of art in ecology and in statistics). Perhaps “stochastic PCR sampling” or something?
2. Lines 79-80: be specific. Rare OTUs are going to be most subject to stochastic amplification/detection.
3. Line 133: wording is misleading. You mean that, within a PCR reaction, forward and reverse PCR primers had identical indices. As written, it sounds as if the same index primers were used for all PCR reactions (which isn’t true).
4. Line 137: it is worth noting that pooling the PCRs for sequencing obscures even greater effects of stochastic amplification that might have been observed if you had sequenced each PCR reaction individually.
5. Line 202 and throughout: you don’t need to refer to Jaccard / B-C distances as “pseudo-beta diversity.” These are perfectly acceptable measures of community (dis-)similarity. I suggest removing the “pseudo-beta diversity” descriptors, simply because I find them confusing (but I may be alone there).
6. Line 203: note that if you are using R (vegan) to calculate Jaccard, it doesn’t work precisely as you’ve described. See ?vegdist(). It seems Jaccard is derived from Bray-Curtis in this implementation, and so you need to use the binary = T option in order for it to work as you have described in the paper.
7. Lines 206-227 is where I think the PERMANOVA framework would really simplify things.
8. Lines 319-320 largely repeat the information on lines 299-300.
9. Lines 395-397: this depends upon the origin of the reads, I think. One can imagine lab-contaminated samples (say, 50% human, 25% mouse, and 25% actual environmental sample) being highly repeatable across replicates if the source sample is contaminated. On the other hand, rare “contaminants” might not be contaminants in that same sense… this gets to a larger conversation about what we might consider “contamination” (or alternatively, “false-postives”) and what our “true” reference community truly contains. But I digress.
10. Figure 3 is excellent and very instructive.
11. Note in Figure 5 that we might have more confidence in rare OTUs that occur in a greater number of replicates. This is the intuition behind occupancy modeling (see Lahoz-Monfort et al 2015 for application in the eDNA context).
12. Figure 6 is also very cool.

·

Basic reporting

No Comments

Experimental design

No Comments

Validity of the findings

No Comments

Additional comments

In this manuscript, Leray & Knowlton looked at the reproducibility of PCR-based metabarcoding studies on a MiSeq platform. By running replicates of PCR and sequencing runs they were able to show that rare OTUs were dropping out of sequencing run replicates, and read abundance could vary by orders of magnitude according to the indexed PCR primers that were used.

I thought the paper was well executed and easy to read, and that the results will be of interest to all researchers working with metabarcoding. I have little to add, but I wondered if it were possible to demonstrate if some of the indexed primer combinations were better than others, i.e. figure 4C shows that P2 and P3 are perhaps the more different; maybe there are some taxon-specific primer issues that could be identified?

The authors may wish to also consider the following very minor suggestions:

L186: You ran BLAST on a local copy of BOLD, or you used the “BOLD Identification System”, which technically isn’t BLAST.

L408: Is “minor” the right word here?

Figure 4B: Could make the bars a big fatter?

Figure 6 legend: “do have any OTU in common” should read “do not have any OTU in common”?

---

## Round 0.2 · accepted · Accept

Dear Matthieu and Nancy,

Thank you for submitting a fine MS to PeerJ. I am happy with how you implemented the reviewer’s comments and suggestions, and thus I am happy to recommend acceptance.

While reading your MS, I found an issue that you will need to correct before the MS is published.

1) Check your statement on line 409-12 (PERMANOVA analysis confirmed that indicating that much of the variation in read abundance was driven by indexed PCRs (R2=98.3%, F(1,6) = 1.059, p = 0.367) but differences were also significant between Illumina runs (R2=0.7%, F(1,6) = 1.049, p = 0.414)) - you are reporting results from Jaccard (non significant), but I think you want to be reporting Bray-Curtis (significant).

I also highly appreciate that you provided a complete set of scripts to accompany this study.

Sincerely,

Tomas Hrbek